# How do agents invest strategically under persistent improvement?

## Abstract

This paper studies algorithmic decision-making under human's strategic behavior, where a decision-maker uses an algorithm to make decisions about human agents, and the latter with information about the algorithm may exert effort strategically and improve to receive favorable decisions. Unlike prior works that assume agents benefit from their efforts immediately, we consider realistic scenarios where the impacts of these efforts are persistent and agents benefit from efforts by making improvements gradually. We first develop a dynamic model to characterize persistent improvements and based on this construct a Stackelberg game to model the interplay between agents and the decision-maker. We analytically characterize the equilibrium strategies and identify conditions under which agents have incentives to improve. With the dynamics, we then study how the decision-maker can design an optimal policy to incentivize the largest improvements inside the agent population. We also extend the model to settings where (1) agents may be dishonest and game the algorithm into making favorable but erroneous decisions; (2) honest efforts are forgettable and not sufficient to guarantee persistent improvements. With the extended models, we further examine conditions under which agents prefer honest efforts over dishonest behavior and the impacts of forgettable efforts.

## 1 Introduction

In applications such as lending, college admission, hiring, etc., machine learning (ML) algorithms have been increasingly used to evaluate and make decisions about human agents. Given information about an algorithm, agents subject to ML decisions may behave strategically to receive favorable decisions. How to characterize the strategic interplay between algorithmic decisions and agents, and analyze the impacts they each have on the other, are of great importance but challenging.

This paper studies algorithmic decision-making under strategic agent behavior. Specifically, we consider a decision-maker who assesses a group of agents and aims to accept those that are *qualified* for certain tasks based on assessment outcomes. With knowledge of the acceptance rule, agents may behave strategically to increase their chances of getting accepted. For example, agents may invest to genuinely *improve* their qualifications (i.e., honest effort), or they may *manipulate* the observable assessment outcomes to game the algorithm (i.e., dishonest effort). Both types of behaviors have been studied. In particular, Hardt et al. (2016a); Dong et al. (2018); Braverman & Garg (2020); Jagadeesan et al. (2021); Sundaram et al. (2021); Zhang et al. (2022); Eilat et al. (2022) focus on learning under strategic manipulation, where they proposed various analytical frameworks (e.g., Stackelberg games) to model manipulative behavior, and analyzed analytical models or developed learning algorithms that are robust against manipulation.

Another line of research (Zhang et al., 2020; Harris et al., 2021; Bechavod et al., 2022; Kleinberg & Raghavan, 2020; Chen et al., 2020; Barsotti et al., 2022; Jin et al., 2022) considers a different setting where agent qualifications (labels) change in accordance with the improvement actions. The goal of the decision-maker is to design a mechanism such that agents are incentivized to behave toward directions that improve the underlying qualifications. The mixture of both improvement and manipulation behavior is also studied (Miller et al., 2020; Chen et al., 2020; Barsotti et al., 2022; Horowitz & Rosenfeld, 2023). However, these existing works regarded improvement as a similar action to manipulation where the only difference is it will incur a label change. Another related topic is *performative prediction* (Perdomo et al., 2020; Izzo et al., 2021; Hardt et al., 2022), an abstraction

that captures agent actions via model-induced distribution shifts. Details and more related works are presented in Appendix C.

This paper primarily focuses on honest agents with improvement, while settings with both improvement and manipulation are also studied. We first propose a novel two-stage Stackelberg game to model the strategic interactions between decision-maker and agents, i.e., the decision-maker commits to its policy, following which agents best respond. A crucial difference between this study and the prior works is that the existing models all assume that the results of agents' improvement actions are *immediate*, i.e., once agents decide to improve, they experience *sudden* changes in qualifications and receive the return *at once*. However, we observe that in many real-world applications, the impacts of improvement action are indeed *persistent* and *delayed*. For example, humans improve their abilities by acquiring new knowledge, but they make progress gradually and benefit from such behavior throughout their lifetime; loan applicants improve their credit behaviors by repaying all the debt in time, but there is a time lag between such behaviors and the increase in their credit scores. Therefore, it is critical to capture these delayed outcomes in the Stackelberg game formulation.

To this end, we propose a *qualification dynamic* model to characterize how agent qualifications would gradually improve upon exerting honest efforts. Such dynamics further indicate the time it takes for agents to reach the targeted qualifications that are just enough for them to be accepted. The impacts of such time lag on agents are then captured by a *discounted utility model*, i.e., reward an agent receives from the acceptance diminishes as time lag increases. Under this discounted utility model, agents best respond by determining how much effort to exert that maximizes their discounted utilities.

This paper aims to analytically and empirically study the proposed model. With the understanding of the strategic interactions between the decision-maker and agents, we further study how the decision-maker can design an optimal policy to incentivize the largest improvements inside the agent population, and empirically verify the benefits of the optimal policy.

Additionally, we extend the model to more complex settings where (i) agents have an additional option of strategic manipulation and can exert dishonest effort to game the algorithm; (ii) honest efforts exerted by agents are forgettable and may not be sufficient to guarantee persistent improvements, instead the qualifications may deteriorate back to the initial states. We will propose a *model with both manipulation & improvement* and a *forgetting mechanism* to study these settings, respectively. We aim to examine how agents would behave when they have both options of manipulation and improvement, under what conditions they prefer improvement over manipulation, and how the forgetting mechanism affects an agent's behavior and long-term qualifications.

Our contributions can be summarized as follows:

1. We formulate a new Stackelberg game to model the interactions between decision-maker and strategic agents. To the best of our knowledge, this is the first work capturing the delayed and persistent impacts of agents' improvement behavior (Sec. 2).
2. We study the impacts of acceptance policy and the external environment on agents, and identify conditions under which agents have incentives to exert honest efforts. This provides guidance on designing incentive mechanisms to encourage agents to improve (Sec. 3).
3. We characterize the optimal policy for the decision-maker that incentivizes the agents to improve (Sec. 4).
4. We consider the possibility of dishonest behavior and propose a *model with both improvement and manipulation*; we identify conditions when agents prefer one behavior over the other (Sec. 5).
5. We propose a *forgetting mechanism* to examine what happens when honest efforts are not sufficient to guarantee persistent improvement (Sec. 6).
6. We conduct experiments on real-world data to evaluate the analytical model and results (Sec. 7).

## 2 PROBLEM FORMULATION

Consider an agent population with $m$ skill sets. Each agent has a *qualification profile* at time $t$, denoted as a unit $m$-dimensional vector $q_t \in [0,1]^m$ with $||q_t||_2 = 1$, A decision-maker at each time makes decisions $D_t \in \{0,1\}$ ("0" being reject and "1" accept) about the agents based on their qualification profiles. Let fixed vector $d \in [0,1]^m$ be the ideal qualification profile that the decision maker desires.

**Decision-maker's policy.** For an agent with qualification profile $q_t$, the decision-maker assesses whether the agent's profile lines up with the desired qualifications $d$, and makes decision $D_t$ based on their similarity $x_t := q_t^T d$ using a *fixed* threshold policy $\pi(x_t) = \mathbf{1}(x_t \geqslant \theta)$, i.e., only agents that are sufficiently fit can get accepted. How to choose threshold $\theta$ is discussed in Sec. 4. We assume only agents with initial similarity $x_0 \geqslant 0$ are interested in positions and only focus on these candidates.

Although the decision policy introduced above focuses on the similarity between $q_t$ and $d$ where qualifications $q_t$ are normalized with the same magnitude for all agents, it can be easily extended to settings where the magnitude/strength of skills also matters and may differ across agents. Specifically, we propose a ***pre-normalization procedure*** to account for the strength of skills. The idea is to first add an additional dimension to initial qualification profile $q_0$, which represents the agent's unobservable "irrelevant attribute" (all other skills an agent has that are not important for the decision). Meanwhile, we add this dimension to ideal qualification profile $d$ with 0 as its value. We can make a natural assumption that *after adding the dimension of "irrelevant attribute", the norm of $(m + 1)$-dimensional complete profile is the same for all agents.* This is reasonable and supported by the literature (Liu et al., 2022; Holmstrom & Milgrom, 1991), which suggests that when qualification profiles are multi-dimensional, the competency in relevant/measurable attributes implies the weakness in irrelevant/unmeasurable attributes. The detailed pre-normalization procedure is formally presented in Algorithm 1 (App. A).

**Agent qualification dynamics.** We assume agents have information about the ideal profile $d$ (e.g., from application guides, mock interviews). In the beginning, agents with $q_0$ can choose to improve their profiles by investing a *one-time* effort $k \in [0, 1]$ to acquire the relevant knowledge, but the effort will have delayed and persistent effects. The specific value of $k$ depends on the agent's utility and will be introduced at the end of this section. Upon making investment $k$, the agent's qualifications $q_t$ gradually improve over time based on the following[1]:

$$\widetilde{q}_{t+1} = q_t + k \cdot q_t^T d \cdot d \, ; \qquad q_{t+1} = \frac{\widetilde{q}_{t+1}}{\|\widetilde{q}_{t+1}\|_2}. \tag{1}$$

equation 1 suggests that agents at each time improve toward the ideal profile $d$. How much they can improve depend on their current profile $q_t$ and the effort $k$. The similarity $q_t^T d$ in the dynamics captures the reinforcing effects: agents that are more qualified could have more resources and are more capable of leveraging the acquired knowledge to improve their skills. Note that the maximum improvement an agent attains at each round is bounded, i.e., the normalized vector $q_{t+1}$ after improvement is always between current qualifications $q_t$ and the ideal profile $d$. Fig. 1 illustrates the improvement dynamics of qualification $q_t$ in a two-dimensional space.

Dynamics in equation 1 model the delayed and persistent impacts of improvement action (i.e., effort $k$). In many real applications, humans acquire knowledge and benefit from repeated practices. They make progress toward the goal gradually, and it takes time to receive the desired outcome from the investment. Indeed, equation 1 is inspired by the dynamics in Dean & Morgenstern (2022), which was used to model the preference shifts of consumers in recommendation systems (details are in App. C.3). We consider a different problem and use the dynamics to model the evolution of agent's (pre-normalized) qualifications. Based on Dean & Morgenstern (2022), we know that $q_t$ converges under dynamics, as formally stated in Lemma 2.1 below.

Figure 1: Qualification dynamics

**Lemma 2.1** (Convergence of qualification). *Consider an agent with initial similarity $x_0 := q_0^T d > 0$. If he/she makes an effort $k$ and improves qualification profile $q_t$ based on dynamics in equation 1, then $q_t$ converges to the desired profile $d$. The evolution of the similarity $x_t := q_t^T d$ is given by:*

$$x_t^{-2} - 1 = \frac{(x_0)^{-2} - 1}{(k + 1)^{2t}} \tag{2}$$

Lemma 2.1 suggests that any agent eventually becomes an ideal candidate with a perfectly aligned profile (i.e., $x_t = q_t^T d = 1$), as long as he/she is interested in the position ($x_0 \geqslant 0$) and willing to make an effort ($k > 0$). The only difference among agents is the speed of convergence: it takes less time for agents who are more qualified at the beginning (i.e., larger $x_0$) and/or make more effort (i.e., larger $k$) to become ideal and get accepted.

---

[1] Though $k$ never changes in equation 1, we relax the design in App. B with convergence results.

**Agent's utility & action.** Because it takes time for agents to receive rewards (i.e., get accepted) for their efforts, they may not have incentives to invest if there is a long delay. In practice, people may be more attracted to investments with immediate rewards than delayed rewards, or they may simply not have enough time to wait. For example, students only have limited time to prepare for college applications; credit card applicants may not have incentives to improve their credit scores and wait to get approval for a specific credit card when there are many instant-approval cards on the market.

To characterize the delayed rewards, we use a discount model and assume the reward each agent receives from the effort $k$ decreases over time. Specifically, let $T$ be the minimum time it takes for an agent to get accepted from the effort $k > 0$. We define **agent's utility** as:

$$U = \frac{1}{(1+r)^T} - k. \tag{3}$$

That is, the utility is the exponentially discounted reward an agent receives from the acceptance minus the effort. $r > 0$ is the discounting factor. Note that the discounted utility model [2] has been widely used in literature such as reinforcement learning (Kaelbling et al., 1996), finance (Meier & Sprenger, 2013), and economics (Krahn & Gafni, 1993; Samuelson, 1937).

Since threshold policy $\pi(x_t) = \mathbf{1}(x_t \geqslant \theta)$ is used to make decisions, an agent gets accepted whenever the qualification profile is sufficiently aligned with the ideal profile, i.e., $x_t = q_t^T d \geqslant \theta$. Based on equation 2, we can derive $T$ as a function of threshold $\theta$, agent's initial similarity $x_0$, and effort $k$, i.e.,

$$T = \min_t \{x_t \geqslant \theta\} = \min_t \left\{ \frac{(x_0)^{-2} - 1}{(k+1)^{2t}} \leqslant \frac{1}{\theta^2} - 1 \right\} = \frac{-\ln\left(\sqrt{\frac{(\theta)^{-2}-1}{(x_0)^{-2}-1}}\right)}{\ln(k+1)} \tag{4}$$

Plug in equation 3, agent's utility becomes:

$$U := U(k, \theta, r, x_0) = (1+r)^{\frac{\ln\left(\sqrt{\frac{(\theta)^{-2}-1}{(x_0)^{-2}-1}}\right)}{\ln(k+1)}} - k. \tag{5}$$

Therefore, strategic agents will choose to improve their qualifications only if utility $U(k, \theta, r, x_0) > 0$, and they will choose the investment $k$ that maximizes the utility.

**Stackelberg game.** We model the strategic interplay between the decision-maker and agents as a Stackelberg game, which consists of two stages: (i) the decision-maker first publishes the optimal acceptance threshold $\theta$ (details are in Sec. 4); (ii) agents after observing the threshold take actions to maximize their utilities as given in equation 5.

**Manipulation & forgetting.** The model formulated above has two implicit assumptions: (i) agents are honest and they improve their qualifications by making actual efforts; (ii) once agents make a one-time effort $k$ to acquire the knowledge, they never forget and can repeatedly leverage this knowledge to improve their profiles based on equation 1. However, these assumptions may not hold. In practice, agents may fool the decision-maker by directly manipulating $x_t$ to get accepted without improving actual $q_t$, e.g., people cheat on exams or interviews to get accepted. Moreover, the knowledge agents acquired at the beginning may not be sufficient to ensure repeated improvements.

Therefore, we further extend the above model to two settings:

1. *Manipulation:* Besides improving the actual profile $q_t$ by making an effort $k$, agents may choose to manipulate $x_t$ directly to fool the decision-maker. The detailed model and analysis are in Sec. 5.
2. *Forgetting:* One-time investment $k$ may not guarantee the improvements all the time, i.e., qualifications $q_t$ do not always move toward the direction of ideal profile $d$, instead it may devolve and possibly go back to starting state $q_0$. The detailed model and analysis are in Sec. 6.

**Objective.** In this paper, we study the above interactions between decision-maker and agents. We aim to understand (i) under what conditions agents have incentives to improve their qualifications; (ii) how to design the optimal policy to incentivize the largest improvements inside the agent population; (iii) how the agents would behave when they have both options of manipulation and improvement, and under what conditions agents prefer improvement over manipulation; (iv) how the forgetting mechanism affects agent's behavior and long-term qualifications.

---

[2]Under exponential discounting function, the agent's reward diminishes at a constant rate (Grüne-Yanoff, 2015). Our model can also adopt other discounting functions (e.g., hyperbolic discounting) for settings when the agent's reward decreases inconsistently. The qualitative results of this paper still remain the same.

## 3   IMPROVEMENT & OPTIMAL EFFORT

In this section, we examine the impact of decision threshold $\theta$ and the environment (i.e., discounting factor $r$) on agent behavior. Specifically, we focus on agents with discounted utility (equation 5) and identify conditions under which the agents have incentives to improve their qualifications. Note that we do not consider issues of manipulation and forgetting in this section. Based on equation 5, an agent with $x_0 := q_0^T d$ chooses to improve only if its utility $U(k, \theta, r, x_0) > 0$. To characterize the impact of an agent's one-time investment $k$ on $U(k, \theta, r, x_0)$, we first define a function $C(\theta, r, x_0)$ that summarizes the impacts of all the other factors (i.e., threshold $\theta$, discounting factor $r$, and initial profile similarity $x_0$) on agent utility, as defined below.

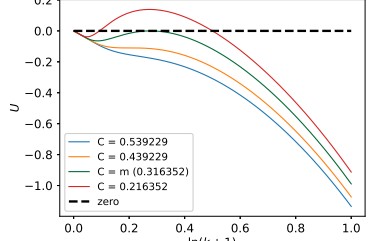

$$C(\theta, r, x_0) = -\ln\left(\sqrt{\frac{(\theta)^{-2} - 1}{(x_0)^{-2} - 1}}\right) \cdot \ln(1 + r) \quad (6)$$

Based on $C(\theta, r, x_0)$, we can derive conditions under which agents have incentives to improve (Thm. 3.1).

**Theorem 3.1** (Improvement & optimal effort). *There exists a threshold $m > 0$ such that for any $\theta, r, x_0$ that satisfies $C(\theta, r, x_0) < m$, the agent has the incentive to improve the qualifications, i.e., agent utility is positive for some efforts $k > 0$. Moreover, there exists a unique optimal effort $k^* \in (0, 1)$ that maximizes the agent utility.*

Figure 2: Impact of effort $k$ on agent utility $U$ under different $C := C(\theta, r, x_0)$: there exists $m > 0$ such that agents have incentives to invest efforts if $C < m$.

Thm. 3.1 identifies a condition under which agents have incentives to exert positive effort $k > 0$. This condition depends on factors $\theta, r, x_0$ and can be fully characterized by the function $C := C(\theta, r, x_0)$.

Although the analytical solution of the threshold $m$ is difficult to find, we can numerically solve $m \approx 0.3164$ as shown in App. H.1. In Fig. 2, we illustrate agent utilities $U$ as functions of effort under different $C$. The results show that only when $C < m$ (red curve), an agent can attain positive utility with effort $k > 0$; when $C \geqslant m$ (green/yellow/blue curve), agents will not invest because the maximum utility is attained at $k = 0$. Moreover, when $C < m$ (red curve), there is a unique optimal effort $k$ that maximizes the utility. These results are consistent with Thm. 3.1.

The condition in Thm. 3.1 further indicates the impacts of policy $\theta$, discounting factor $r$, and initial state $x_0$ on agent behavior. Specifically, agents only invest if $C(\theta, x_0, r) < m$ holds. By fixing any two of $\theta, x_0, r$, we can identify the domain of the third factor

Table 1: Domain of initial similarity $x_0$ (or threshold $\theta$) under which agents invest positive efforts, given other factors fixed.

| | |
|---|---|
| **Domain of $x_0$ (given $\theta, r$)** | $x_0 > \left(1 + (\theta^{-2} - 1) \cdot \exp\left(\frac{2m}{\ln(1+r)}\right)\right)^{-1/2}$ |
| **Domain of $\theta$ (given $x_0, r$)** | $\theta \leqslant \left(1 + (x_0^{-2} - 1) \cdot \exp\left(-\frac{2m}{\ln(1+r)}\right)\right)^{-1/2}$ |

under which agents invest to improve. These results are summarized in Table 1 and verified in App. D. It shows that for any threshold $\theta$ and discounting factor $r$, agents only improve if their initial qualification profile is sufficiently similar to the ideal profile; the domain of $\theta$ also implies the best profile an agent with initial state $x_0$ can reach after exerting effort: if acceptance threshold $\theta$ is larger than the upper bound of $\theta$ given in Table 1, then agents will not have incentives to improve.

The above results further suggest effective strategies that encourage agents to improve their qualifications, i.e., more agents are incentivized to improve if (i) the decision-maker's acceptance threshold $\theta$ is lower; or (ii) the time it takes for agents to succeed after investments is shorter (smaller discounting factor $r$). Examples of both strategies in real applications are discussed in App. D.

## 4   DECISION-MAKER'S POLICY TO INCENTIVIZE IMPROVEMENT

Sec. 3 studied the impact of threshold $\theta$ on agent behavior and provided guidance on incentivizing agents to improve. In practice, although it is more difficult to adjust the discounting factor $r$, the decision-maker can adjust the threshold policy $\theta$ to incentivize the largest possible amount of total improvement, thereby improving the *social welfare*. In this section, we study the optimal policy when the decision-maker is aware of the agent's best response and hopes to incentivize agents to improve.

Suppose the decision-maker has full information about agents and can anticipate their behaviors, i.e., for any decision threshold $\theta$, it knows that agents whose initial similarity $x_0 > x^*(\theta) := \left(1 + (\theta^{-2} - 1) \cdot \exp\left(\frac{2m}{\ln(1+r)}\right)\right)^{-\frac{1}{2}}$ will invest and improve their profiles (by Table 1). Also, we define $x^*(0) = 0$ to let $x^*(\theta)$ be continuous in $[0,1]$ and denote $f$ as the probability density function of the agent similarity $x_0$ which is also continuous in $[0,1]$. Then, we can define $U_d(\theta)$ as the utility of the decision-maker under the threshold as the total amount of agents' improvements as follows.

$$U_d(\theta) = \int_{x^*(\theta)}^{\theta} (\theta - x_0) \cdot f(x_0)dx_0 \tag{7}$$

The above equation 7 demonstrates that the decision-maker aims to maximize the total improvement among the agent population, and its utility is a function of $\theta$. Since $f(x), x^*(\theta)$ are both continuous in $[0,1]$, utility $U_d(\theta)$ is also continuous. The following Thm. 4.1 further shows the existence of the optimal thresholds $\theta^* \in (0,1)$.

**Theorem 4.1** (Existence of optimal threshold). *For any decision-maker with utility function $U_d$, there exists at least one $\theta^* \in (0,1)$ that is optimal under which $U_d(\theta) > 0$. Moreover, $\theta^*$ is the unique optimal point of $U_d$ if $\frac{\partial U_d}{\partial \theta}$ has one root within $(0,1)$.*

To verify Thm. 4.1, we demonstrate the values of $U_d$ under situations where the agent population has different density functions $f$ and different discounting factors $r$. Specifically, we consider the uniform distribution and Beta distributions with different parameters. Fig. 3 shows $U_d(\theta)$ under different density functions $f$ and discounting factors $r$. The results illustrate that under these settings, $U_d$ is single-peaked and there is a unique $\theta^* \in (0,1)$ that is optimal and results in positive utility, which is consistent with Thm. 4.1. The figure also indicates the impact of $r$ on the optimal threshold: as $r$ increases, $\theta^*$ increases and the corresponding maximum utility decreases. As formally stated below in Corollary 4.2. We prove Thm. 4.1 and Corollary 4.2 in App. H.2.

**Corollary 4.2.** *For $U_d(\theta)$ that has a unique maximizer $\theta^*$, optimal $\theta^*$ decreases as $r$ increases.*

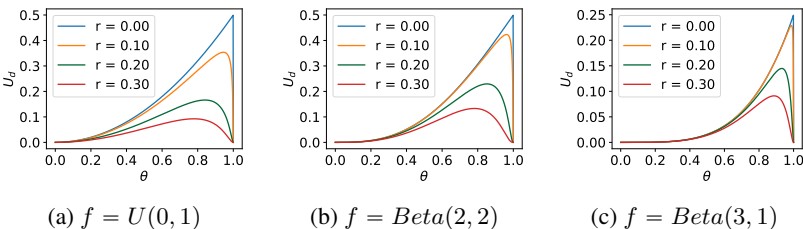

(a) $f = U(0,1)$       (b) $f = Beta(2,2)$       (c) $f = Beta(3,1)$

Figure 3: Optimal thresholds $\theta^*$ under different density functions $f$ and discounting factors $r$.

Importantly, the results of Thm. 4.1 show that the decision-maker can always find an optimal decision threshold $\theta^*$ (either numerically or using gradient methods depending on the density function $f$) to incentivize the largest improvement and promote *social welfare* in practice. While the above results all assume the decision-maker knows $r$ when determining $\theta$, we can relax this and provide a procedure to estimate $r$; this is included in App. G.

## 5   IMPACT OF MANIPULATIVE BEHAVIOR

Our analysis and results so far rely on an implicit assumption that agents are honest and they improve qualifications $q_t$ by making actual efforts. However, as mentioned in Sec. 2, agents in practice may fool the decision-maker by strategically manipulating $x_t = q_t^T d$ to get accepted without improving $q_t$. Next, we extend our model in Sec. 2 by considering the possibility of such manipulative behavior.

**Model with both manipulation & improvement.** We extend the model in Sec. 2 where agents after observing $\theta$ have an additional option to manipulate $x_0$ directly. If they choose to ***improve***, they make a one-time effort $k \in [0,1]$ to acquire relevant knowledge and gradually improve their qualifications $q_t$ over time based on equation 1. If they choose to ***manipulate***, they only increase $x_t$ at every round to fool the decision-maker without changing the actual profile $q_t$. Similar to the literature on strategic classification (Hardt et al., 2016a), the manipulation comes at the cost and the risk of being caught.

Specifically, let $c(x', x) \geqslant 0$ be the *manipulation cost* it takes for an agent to increase its similarity from $x$ to $x'$, and $P \in [0, 1]$ be the *detection probability* of manipulation during an agent's entire application process. Agents, once getting caught manipulating $x_t$, will never be accepted.

**Degree of manipulation.** If agents choose to manipulate, they will increase $x_t$ at every round to fool the decision-maker, and they manipulate in a way that minimizes the manipulation cost and the risk of being detected. We make the following natural assumptions on $c$ and $P$:

1. Let $\overline{x}_t$ be the best outcome agents can attain from $x_{t-1}$ at round $t$ by improvement behavior (with largest effort $k = 1$). If $x_t > \overline{x}_t$ for some $t$, then $P = 1$ because the decision-maker can be certain that $x_t$ is the result of manipulation; otherwise, $P \in [0, 1)$ if $x_t \leqslant \overline{x}_t$.

2. The total manipulation cost it takes for an agent with initial similarity $x_0$ to be accepted is $c(\theta, x_0)$.

Note that $\overline{x}_t$ above indicates the maximum degree of manipulation of agents: to avoid being detected, an agent should not manipulate $x_t$ more than $\overline{x}_t$. We can compute $\overline{x}_t$ directly from Lemma 2.1 (by setting $k = 1$), i.e., $\overline{x}_t = \left( \frac{x_{t-1}^{-2} - 1}{4} + 1 \right)^{-\frac{1}{2}}$. For agents who manipulate, if the total manipulation cost needed to get accepted is $c(\theta, x_0)$ and detection probability $P = 1$ whenever $x_t > \overline{x}_t$, then agents will always manipulate toward $\overline{x}_t$ to maximize its utility. As a result, agents who manipulate can be regarded as they mimic the improvement behavior with the largest effort $k = 1$.

Let $\widetilde{U}$ be agent's **utility under manipulation**, which is the benefit an agent obtains from acceptance (when not being detected) minus the manipulation cost, i.e.,

$$\widetilde{U} = (1 - P) \cdot (1 + r)^{\frac{-\ln\left(\sqrt{\frac{(\theta)^{-2} - 1}{(x_0)^{-2} - 1}}\right)}{\ln 2}} - c(\theta, x_0), \tag{8}$$

where the benefit is derived based on equation 5 (with $k = 1$).

**Agent's best response.** Suppose agents have full information about detection probability $P$ and discounting factor $r$, after observing the acceptance threshold $\theta$, they best respond by choosing the action (i.e., improvement/manipulation/do nothing) that maximizes their utilities, i.e., if $\widetilde{U} > \max_k U$, they choose to manipulate; otherwise, they improve by exerting optimal effort $k^* = \arg\max_k U$.

Next, we examine under what conditions agents prefer improvement over manipulation.

**Theorem 5.1.** *Suppose manipulation cost* $c(x', x) = (x' - x)_+$ *and threshold* $\theta \geqslant \overline{\theta}$ *for some* $\overline{\theta} \in (0, 1)$. *For any discounting factor* $r$, *there exists* $\widehat{P} \in (0, 1)$ *such that the followings hold:*

*1. If* $P = 0$, *then* $\exists \widehat{x} \in (0, 1)$ *such that agents manipulate only when initial similarity* $x_0 \in (\widehat{x}, \theta)$.

*2. If* $P \in (0, \widehat{P}]$, *then* $\exists \widehat{x_1}, \widehat{x_2}$ *such that agents manipulate only when initial* $x_0 \in (\widehat{x_1}, \widehat{x_2})$.

*3. If* $P > \widehat{P}$, *then agents never choose to manipulate.*

Thm. 5.1 considers scenarios when the threshold is sufficiently high, and identifies conditions under which manipulation is preferred by agents in these settings. It shows agent behavior highly depends on the risk of manipulation (i.e., detection probability $P$). The specific values of $\widehat{P}, \widehat{x}, \widehat{x_1}, \widehat{x_2}$ in Thm. 5.1 depend on $\theta, r$. In particular, $\widehat{P}$ increases as $r$ increases. Indeed, we can empirically find $\widehat{P}, \widehat{x}, \widehat{x_1}, \widehat{x_2}$ and verify the theorem. These are illustrated in App. E and Sec. 7.

# 6 FORGETTING MECHANISM

The analysis in previous sections relies on the assumption that once agents make a one-time effort $k$ to acquire the knowledge, they never forget and can repeatedly leverage this knowledge to improve their profiles based on equation 1. This may not hold in practice when the knowledge agents acquired at the beginning are not sufficient to guarantee repeated improvements. In this section, we extend the qualification dynamics (equation 1) by incorporating the *forgetting mechanism*, i.e., qualification profile $q_t$ does not always move toward the direction of ideal profile $d$, instead, it may devolve and possibly go back to the initial $q_0$. Note that we only consider honest agents who do not manipulate. By modifying equation 1, we define the new **qualification dynamics with forgetting** as follows.

$$\widetilde{q}_{t+1} = q_t + (k \cdot d + (1 - k) \cdot q_0) \cdot q_t^T d; \qquad q_{t+1} = \frac{\widetilde{q}_{t+1}}{\|\widetilde{q}_{t+1}\|_2} \tag{9}$$

Let $\widetilde{d} := k \cdot d + (1-k) \cdot q_0$, then new dynamics in equation 9 implies that at each round, qualification profile $q_t$ is pushed toward the direction of $\widetilde{d}$, i.e., a convex combination of ideal profile $d$ and initial qualifications $q_0$. Whether $q_t$ improves towards $d$ or deteriorates back to $q_0$ depends on the investment $k$: with more effort $k$, the degree of forgetting is less; there is no forgetting if all the knowledge is acquired ($k = 1$). Under the new dynamics, we can derive the convergence of the qualification profile as follows.

**Theorem 6.1** (Convergence of qualification under forgetting). *Consider an agent with initial similarity $x_0 = q_0^T d > 0$ whose qualifications $q_t$ follow dynamics in equation 9. Suppose the agent makes investment $k > 0$, then $q_t$ converges to profile d\* and the similarity $x_t = q_t^T d$ satisfies:*

$$(x_t^*)^{-2} - 1 < \frac{(x_0^*)^{-2} - 1}{(k_u + 1)^{2t}} \tag{10}$$

*where $d^* = \frac{\widetilde{d}}{\|\widetilde{d}\|}$, $x_t^* = q_t^T d^*$, and $k_u = \|\widetilde{d}\| \cdot x_0$.*

Thm. 6.1 implies that convergence still holds when qualifications evolve with forgetting. Unlike the scenarios without forgetting where $q_t$ eventually converges to the ideal profile $d$ regardless of $k$ (Lemma 2.1), $q_t$ now converges to $d^*$, i.e., a profile between initial qualifications $q_0$ and ideal profile $d$, which is closer to $q_0$ with smaller investment $k$. It shows that if agents do not exert enough effort and the acquired knowledge is not sufficient, then they will not make satisfactory improvements.

**Agent's utility and improvement action.** Denote agent utility under the forgetting mechanism as $\widehat{U}(k, \theta, r, x_0)$. Unlike settings without forgetting where we can derive the exact time $T$ it takes for agents to be accepted and find utility $U$ (equation 5), the analytical form of $\widehat{U}(k, \theta, r, x_0)$ is not easy to derive. Nonetheless, we can still show that there exist scenarios under which agents have incentives to improve, even though the best attainable profile is a profile $d^*$ between initial $q_0$ and the ideal $d$.

**Theorem 6.2.** *For any threshold $\theta$ (resp. discounting factor $r$), there exists a discounting factor $r$ (resp. threshold $\theta$) such that agent's utility $\widehat{U}(\bar{k}, \theta, r, x_0) > 0$ for some $\bar{k} \in (0, \widehat{k})$, i.e., agents have the incentive to make a positive effort. The upper bound of the optimal effort is $\widehat{k}$ given by*

$$\widehat{k} = \min\left( \frac{\widehat{x}_0^2}{2\widehat{x}_0^2 + 2\widehat{x}_0^3}, \frac{x_0 \cdot (x_0^2 + x_0 - \sqrt{x_0^4 - x_0^2 + 1})}{2x_0^2 + 2x_0^3 - 1} \right) \tag{11}$$

*where $\widehat{x}_0$ is the root of $2x_0^2 + 2x_0^3 - 1 = 0$ within $(0, 1)$.*

Thm. 6.2 implies that there exists $(\theta, r)$ such that agents best respond by improving their qualifications, and the optimal effort is upper bounded by $\widehat{k}$. Indeed, we can numerically find the upper bound $\widehat{k}$ as a function $x_0$ (shown in Fig. 4). Because $\widehat{k} < 0.35$ for all $x_0$, the improvement an agent can make under the forgetting mechanism is limited.

*Remark* 6.3. Under the forgetting mechanism, the actual effort invested by any agent is less than 0.35, and the qualifications $q_t$ converge to a profile $d^*$ that is between $q_0$ and $0.35 \cdot d + 0.65 \cdot q_0$.

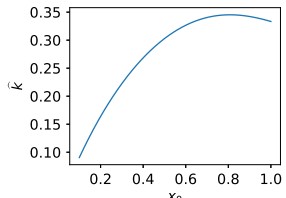

Figure 4: Upper bound $\widehat{k}$ of the optimal effort as a function of $x_0$.

## 7 EXPERIMENTS

We validate theoretical results by conducting experiments on Exam score (Kimmons, 2012) and FICO score (Reserve, 2007) dataset. For both datasets, scores serve as the agent's initial similarity $x_0$, and we assume agents interact with a decision maker based on the Stackelberg game in Sec. 2. We first fit these scores with beta distributions, i.e., $x_0 \sim \text{Beta}(v, w)$, and then use them to derive the followings:

1. The optimal decision threshold $\theta^*$ for the decision-maker to incentivize the largest amount of improvement and promote *social welfare*, and the total improvement induced by $\theta^*$.
2. The percentage of agents who choose to manipulate under the decision-maker's optimal policy.

**Exam Score Data.** It is a synthetic dataset containing 1000 students' exam scores on 3 subjects including math, reading, and writing (Kimmons, 2012). We first average over 3 subjects and normalize

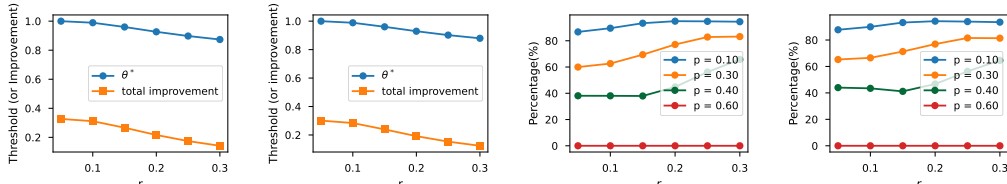

Figure 5: From the left to the right are: optimal thresholds to incentivize improvement for males/females; manipulation probability under the thresholds for males/females for **Exam data**.

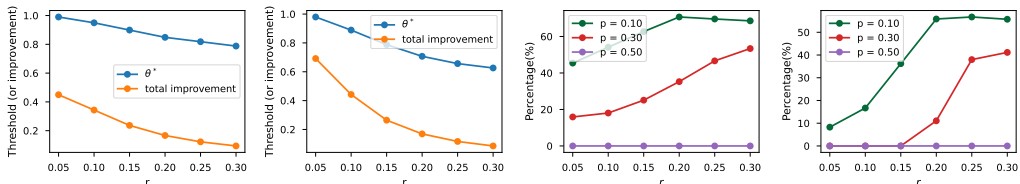

Figure 6: From the left to the right are: optimal thresholds to incentivize improvement for Caucasians and African Americans; manipulation probability under the thresholds for Caucasians and African Americans for **FICO data**.

the averaged score to $[0, 1]$. Then, we fit two beta distributions to the normalized scores of males and females and obtain $x_0 \sim \text{Beta}(4.86, 2.37), \text{Beta}(4.15, 1.79)$ (see Fig. 9 in App. F).

With these distributions, we can compute the optimal decision thresholds and the corresponding total improvement under different discounting factors $r$. As shown in Fig. 5, for both males and females, the experimental results are similar. When $r$ increases, $\theta^*$ always decreases and the total amount of improvement becomes lower. This illustrates how larger discounting factors harm agents' improvement. Additionally, we consider settings with both manipulation and improvement. Fig. 5 also shows the percentages of agents who prefer to manipulate under $\theta^*$. It shows that agents are less likely to manipulate as detection probability $P$ increases.

**FICO Score Data.** We adopt the data pre-processed by Hardt et al. (2016b), which contains CDF of credit scores of four racial groups (Caucasian, African American, Hispanic, Asian). For each group, we fit a Beta distribution and obtain four distributions: $\text{Beta}(1.11, 0.97)$ for Caucasian, $\text{Beta}(0.91, 3.84)$ for African American, $\text{Beta}(0.99, 1.58)$ for Hispanic, $\text{Beta}(1.35, 1.13)$ for Asian (see Fig. 10 in App. F). We only present the results for Caucasians and African Americans, while the results for Asian and Hispanic are shown in App. F.

For each group, we compute the optimal decision threshold and corresponding total improvement under different $r$. As shown in Fig. 6 (left two plots), for both groups, their corresponding optimal threshold $\theta^*$ and the total amount of improvement always decrease as $r$ increases. For settings with both manipulation and improvement (right two plots in Fig. 6), agents are more likely to manipulate under smaller detection probability. When detection probability $P$ is sufficiently large, agents do not have incentives to manipulate.

## 8 SOCIETAL IMPACTS & LIMITATIONS

This paper studies the strategic interactions between agents and a decision-maker when agent action has delayed and persistent effects. Our theoretical results depend on the assumption that both agents and the decision-maker have perfect information about each other so that they always best respond. Extension to cases when each party only has partial/imperfect information is important. Moreover, these theorems are based on the qualification dynamics equation 1. Although a scenario when it does not hold is studied in Sec. 6, future works should also consider other variants tailored to specific applications to prevent negative outcomes. Finally, though we provide a procedure to estimate the discounting factor, performing controlled experiments is not always accessible. Moreover, manipulation cost and detection probability are unknown and hard to estimate. Collecting real data and estimating these parameters remain promising research directions.

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
