# OpenReview forum: "How do agents invest strategically under persistent improvement?"
_ICLR.cc/2024/Conference — ICLR 2024 Conference Withdrawn Submission_

### Official Review · Reviewer_6GuR · 2023-10-31

**Soundness:** 1 poor
**Presentation:** 2 fair
**Contribution:** 1 poor
**Rating:** 1
**Confidence:** 3

**Summary:**

This paper investigates an interesting problem of agents' learning to better satisfy the requirements for being accepted (for completing a task, etc.)  Though the paper emphasises that the objects are human agents, I think the whole paper considers only hypothetical/ artificial agents as the authors do not base the settings/ hypotheses on any verified human data.  They are just reasonable and plausible assumptions and there are no evidence that humans behave exactly in the way as depicted by the hypotheses.

The paper first gives a basic model for agent qualification dynamics (equation 1).  The relation between similarity and time is then given by equation 2.  Utility of an agent is hypothesised in equation 3, from which equation 5 is derived.  Equation 5 is the central equation of the paper as it defines utility U in terms of acceptance threshold theta.  Based on equation 5, Table 1 presents the condition U > 0 in terms x0 and theta. Corollary 4.2 shows that there is an optimal theta that maximises U_d defined in equation 7.  Section 5 changes the setting so that an agent can change its similarity in consideration of manipulation cost and detection probability.  Section 6 changes allows forgetting to happen.  Experimental studies are done in Section 7.

**Strengths:**

The paper is original in some aspect, in particular it captures the delayed and persistent impacts of agents' improvement, as the authors claim.  The quality of writing is good and clear.

The paper's significance is hindered by a possible technical flaw, which I shall elaborate in the weaknesses section below.

**Weaknesses:**

The major question that I have about the paper is the one-time effort k.  The authors emphasise repeatedly that the effort is a one-time effort, which the agent pays at time t=t_i, say.  Such an effort paid changes the agent's qualification from q_{t_i} at time {t_i} to q_{t_{i+1}} at time t_{i+1}, as depicted in equation 1.  From equation 1 the authors derive equation 2.  Now my question is, for equation 2 to hold for any time t, is it not true that the agent needs to pay effort k again at time t_{i+1}, so that its qualifications will be changed from q_{t_{i+1}} to q_{t_{i+2}} according to equation 1?  If what I understand is correct, then equation 3 will need to be revised, so will virtually all subsequent equations and conclusions.

Another issue is that the definition of function C (equation 6) looks arbitrary.  It is unclear to me what C is and why we need this function.  In fact, the definitions of U (equation 3) and U_d (equation 7) are equally arbitrary, but at least the authors explain what thehy are and why they are defined in that way.  I have to point out that though equations 3 and 7 give reasonable definitions for U and U_d, there are no evidence that these *are* the utility functions real people use.

Such an issue is related to another issue (more serious in my view) in section 7.  Basically section 7 only presents a numerical simulation of the relevant equations, instead of proving their correctness in any way.  As a matter of fact, it does not really matter what dataset the authors use--even an artificially created dataset would have served this purpose very well, because only the initial conditions are obtained from the dataset.  Therefore, I do not see the use of this section.  By the way, I do not have the assess to the Exam dataset.

One last point is that the authors use superscript 'T' sometimes to denote 'to the power of T' and sometimes to denote 'transpose'.  This is confusing.

**Questions:**

(1) Is it true that an agent needs to pay effort k in every time step in order for equation 2 to hold?

(2) What is function C and why (and how) is it useful?

(3) Does section 7 only present a numerical simulation of the relevant equations?  What is its significance?

---

### Official Review · Reviewer_m2xs · 2023-11-03

**Soundness:** 3 good
**Presentation:** 3 good
**Contribution:** 2 fair
**Rating:** 5
**Confidence:** 5

**Summary:**

The paper investigated a Stackelberg game between agents and a decision-maker where agents choose the effort to improve their profile to get accepted and the agents maximize the social welfare, which is the total amount of agents' improvements. The authors consider a specific scenario where the effect of agents' action is delayed and has persistent effects. Also, they consider the dishonest behavior of the agents and a forgetting mechanism that captures the case when efforts don't transform to improvements.

**Strengths:**

1. The paper presents the idea and formulation clearly. The organization of the paper is structured in an order that facilitates easy readability and understadability.

2. The study of a Stackelberg game with delayed and persistent impacts of agents' action is important. The problem has not been investigated thoroughly in the context of the specific model presented in this paper.

3. The proof of the theorems and lemmas are well presented and sound.

**Weaknesses:**

1. My major concern is whether the paper fits the scope of the conference or not. It seems to the reviewer that this paper is more suited for a economic journal/conference based on the topic, the way it is presented, and the results. I did not see any discussion in the paper that are related to learning. This is concern drives my rating for this paper.

2. The model formulation is not a generic stackelberg game. The utility function, the dynamics of the model, and the action profile of the agents are defined to study a very specific game. The suggest the authors give a name to the specific game instead of calling it "a novel two-stage Stackelberg model". I was having an impression that the paper studies a generic stackelberg game with delayed and persistent effects when reading the introduction section.

3. The contributions of the results are not significant. For example, the two takeaways from the two theorems in section 6 are 1. the dynamics under forgetting still converges, and 2. the improvement an agent can make under the forgetting mechanism is limited. The two takeaways seem intuitive give the setup of the forgetting mechanism. There are other examples in the previous sections too.

**Questions:**

1. Have the authors considered using dynamic stackelberg game framework to study the scenario, in which agents can change their actions at each time step?

2. Can the authors summarize the most interesting results at the beginning/end of the paper? In section 1, the contributions the authors listed are all about "we formulate", "we study", "we propose". The readers would like to see more insights that we derived from the model and how these results or insights can make an impact in the community.

---

### Official Review · Reviewer_gzfu · 2023-11-08

**Soundness:** 2 fair
**Presentation:** 3 good
**Contribution:** 2 fair
**Rating:** 5
**Confidence:** 2

**Summary:**

This paper considers a problem where an agent has the power to improve or manipulate its feature with delayed effect. The problem is modeled as a Stackelberg game, and the paper studied the optimal strategy in several different settings such as from the agent or the designer, whether manipulation, forgetting is involved.

**Strengths:**

- The model proposed by this paper is indeed novel and interesting to me.
- The paper is overall easy to follow and the figures are helpful to understand the model and experiments.
- The paper used real world data to showcase the use of its model.

**Weaknesses:**

While the model is interesting and could form a reasonable game, I have a major concern about this paper. That is, it is unclear to me how the model is a good reflection of any real world problem. The assumption on the agent qualification dynamics, manipulation/forgetting, and the designer's threshold policy seem to be very strong and artificial. The experiment on real world data also does not reflect whether the modeling is realistic. Hence, I am not sure whether the results derived from this model are particularly useful or insightful for the ICLR audience.

**Questions:**

Can you explain how realistic this model is, and how any of the results derived from this model could be useful for the ICLR community?